# Real-Time Interferometric Refractive Index Change Measurement for the Direct Detection of Enzymatic Reactions and the Determination of Enzyme Kinetics

**DOI:** 10.3390/s19030539

**Published:** 2019-01-28

**Authors:** Søren T. Jepsen, Thomas M. Jørgensen, Henrik S. Sørensen, Søren R. Kristensen

**Affiliations:** 1Department of Clinical Biochemistry, Aalborg University Hospital, Sdr. Skovvej, DK-9000 Aalborg, Denmark; srk@rn.dk; 2Department of Clinical Medicine, Aalborg University, Sdr. Skovvej, DK-9000 Aalborg, Denmark; 3Department of Applied Mathematics and Computer Science, Technical University of Denmark, DK-2800 Kongens Lyngby, Denmark; tmjq@dtu.dk; 4Department of Photonics Engineering, Technical University of Denmark, DK-2800 Kongens Lyngby, Denmark; biz4doc@gmail.com

**Keywords:** refractive index, interferometry, enzyme, *dn*/*dc*

## Abstract

Back scatter interferometry (BSI) is a sensitive method for detecting changes in the bulk refractive index of a solution in a microfluidic system. Here we demonstrate that BSI can be used to directly detect enzymatic reactions and, for the first time, derive kinetic parameters. While many methods in biomedical assays rely on detectable biproducts to produce a signal, direct detection is possible if the substrate or the product exert distinct differences in their specific refractive index so that the total refractive index changes during the enzymatic reaction. In this study, both the conversion of glucose to glucose-6-phosphate, catalyzed by hexokinase, and the conversion of adenosine-triphosphate to adenosine di-phosphate and mono-phosphate, catalyzed by apyrase, were monitored by BSI. When adding hexokinase to glucose solutions containing adenosine-triphosphate, the conversion can be directly followed by BSI, which shows the increasing refractive index and a final plateau corresponding to the particular concentration. From the initial reaction velocities, *K_M_* was found to be 0.33 mM using Michaelis–Menten kinetics. The experiments with apyrase indicate that the refractive index also depends on the presence of various ions that must be taken into account when using this technique. This study clearly demonstrates that measuring changes in the refractive index can be used for the direct determination of substrate concentrations and enzyme kinetics.

## 1. Introduction

The detection and quantification of enzymatic reactions is widely used in many areas of industrial production and biomedical assays. Consequently, there is also a wide range of well-established methods and sensors for measuring such reactions, including but far from limited to colorometric, polariometric and amperometric sensors [1]. Many of these sensors rely on a coupled reaction scheme with secondary reactants that form a detectable biproduct, e.g., NAD^+^/NADH, or enable a transfer of electrons to electrodes to produce a detectable signal. The implementation of such sensors into miniaturized microfluidic systems is therefore challenged by the need to include additional reactants and to ensure the precise mixing of sample and reactant. Furthermore, such coupled reactions may interfere with the reaction rates if secondary products either inhibit the enzyme or have a lower catalytic rate of formation than the primary product, making them less desirable when one wants to study the kinetic parameters of the enzyme itself.

Direct detection can be possible if either the substrate or the product exert distinct physiochemical properties such as chirality, intrinsic fluorescence or in this case, a different refractive index (RI). The refractive index of a dense medium is a macroscopic quantity that is related to the microscopic polarizability through the Lorentz–Lorentz equation:(1)$$\frac{n^{2}-1}{n^{2}+1}=\frac{4\pi }{3}\;N\alpha$$
where *n* is the refractive index, *N* is the number density and *α* is the polarizability. 

Because of the non-invasive nature and ease of miniaturization, RI-based sensors are often used to detect protein and other biomolecules of interest in the emerging area of nano- and microfluidics [2,3] and the possibility of expanding the use of refractive index sensors to also include enzymatic reactions poses an interesting possibility [4,5,6,7].

Previously, we briefly demonstrated that the direct detection of the enzymatic phosphorylation of glucose is possible by measuring the change in the refractive index of a solution, using both an interferometric setup and a conventional differential refractometer [8]. Enzymatic reactions have previously been characterized from RI changes using mass sensitive surface plasmon resonance sensors, however, the enzymes were bound to a solid surface and the change in RI occurred on localized surfaces as a result of mass density changes [9,10]. Likewise, refractometry has been used to study the enzymatic degradation of small micelle-like particles [11]. To the best of our knowledge the characterization of small molecule enzymatic kinetics from RI changes of a bulk fluid has not previously been studied.

In this study, we pursued the use of a refractive index sensor based on so-called back scatter interferometry (BSI), to directly detect and quantify enzymatic reactions in a microfluidic environment. Studies have shown that BSI can detect low concentrations of both ions and saccharides in capillary electrophoresis [12,13,14] and an early study even used a polariometric-BSI configuration to detect an enzymatic reaction with hydroxybutyrate dehydrogenase [15]. There have also been multiple studies on the molecular interactions between protein-sized molecules using BSI [16], but the effect of such interactions on the bulk refractive index and origin of the observed signal has been debated [12,17,18,19]. 

In the present paper, we limited our study to two enzymatic reactions: The first reaction was the phosphorylation of glucose to glucose-6-phosphate (G-6-P) with adenosine-triphosphate (ATP) as a bisubstrate, catalyzed by the enzyme hexokinase, which we have previously shown [8]. In this study, we examined if the observed change in RI during the initial reaction rate was suitable for determining the kinetic Michaelis–Menten constant (*K_M_*). Secondly, given that the former reaction involves multiple products, we sought to differentiate the refractive index contributions of the individual components by studying the enzymatic hydrolysis of adenosine-triphosphate to adenosine di-phosphate and mono-phosphate (ATP, ADP and AMP, respectively) catalyzed by the enzyme apyrase.

Our detection principle was based on the assumption that there is a measurable difference in the refractive index between the solute substrates and products formed in the enzymatic reaction. The observed change in the RI of a solution as a function of solute concentration is termed the refractive index increment (*dn*/*dc*). If the *dn*/*dc* value of the product is different from that of the substrate, one would expect a change in the bulk RI as the reaction proceeds, proportional to the difference in the *dn*/*dc* value between the substrate and the product. We determined the *dn*/*dc* values of ATP and ADP from a series of standard dilutions and discuss these values with the observed RI changes from the enzymatic reactions. The *dn*/*dc* values for glucose and G-6-P have previously been determined using BSI [8].

## 2. Materials and Methods

The back scatter interferometry setup was identical to that used in our previous work [8,19]. Briefly, the beam from a HeNe laser impinges a glass capillary with inner and outer diameters of 1.4 and 1.9 mm respectively, which contains the sample of interest. The probe volume is in itself quite small, less than 5 µL, but the total sample volume injected must be sufficient to ensure full fluid displacement. As light from the laser traverses through the sample liquid, it obtains a phase shift proportional to the RI of the liquid, and as light is reflected from both the inner and outer capillary surfaces, an interference pattern consisting of high contrast periodic fringes is formed. This fringe pattern is recorded with a charge-coupled device and a Fourier transformation is applied to the image. Changes to the sample RI produce a spatial translation of the fringes that is quantified by obtaining the phase of the Fourier transformed image in units of radians (rad).

When observing the directly backwards-reflected fringes the sensitivity of the interferometer (*dϕ*/*dRI*), i.e., the phase response per change in sample RI, is given by the following relationship [20]:(2)$$d\phi /dRI=\left(\frac{2\pi }{\lambda }\right)2nd$$
where *λ* is the laser wavelength (632.8 nm), *n* is the refractive index and *d* is the inner diameter of the capillary (1.4 mm), multiplied by 2 because the backwards reflected light passes the sample twice. In the current setup, this yields a theoretical sensitivity, where *dϕ*/*dRI* = 27,802 rad/RIU. Conversion from radians into refractive index units (RIU) is achieved by the division of the sensitivity. The limit of detection and limiting factors thereof was examined in our previous work and was found to be 7 × 10^−7^ RIU for the current system [19].

The refractive index of a fluid is largely affected by the temperature (10^−4^/°C) and stable temperature control over time is essential for accurately detecting small RI changes during the course of a measured reaction. The current temperature control of the system does not produce baseline deviations greater than 10^−7^ over a period of hours [19]. All measurements were performed at 25 °C.

### 2.1. Reagents

D-Glucose anhydrous of high purity (>99%) was obtained from Merck (Darmstadt, Germany). Glucose-6-phosphate (G7772), hexokinase from *Saccharomyces cerevisiae* (H4502), apyrase from *Solanum tuberosum*, (A6535), adenosine 5′-triphosphate disodium salt hydrate (A2383), and adenosine 5’-diphosphate sodium salt (A2754) were all acquired from Sigma-Aldrich (St. Louis, MO, USA).

### 2.2. Determination of the dn/dc Value

The *dn*/*dc* value was determined using weighed solutions or ATP and ADP prepared in a MES buffer (identical to that used in the enzymatic reactions with apyrase, see below). The *dn*/*dc* value was found by linear regression of the measured signal vs. the concentration.

### 2.3. Measurement of Enzymatic Reactions

Substrate and enzyme mixtures were rapidly mixed by pipetting and 400 μL was injected into the BSI system using a peristaltic pump (BT100-2J, Longer Precision Pump Co., Ltd., Hebei, China). The total mixing and injection time was 40 s, in order to ensure full fluid displacement inside the capillary. The hexokinase measurements were performed using a 1 nM hexokinase solution in a 0.05 M tris buffer, pH 8.0, with 2 mM ATP and 15 mM MgCl_2_. The initial reaction velocity as a function of the substrate concentration (*V_[S]_*) in units of rad/min was measured over a range of glucose substrate concentrations [0.1–2.0 mM] and the Michaelis–Menten constant (*K_M_*) was derived by non-linear parameter estimation using the equation:(3)$$V_{\left[S\right]}=V_{max}\left(\left[S\right]/\left[S\right]+K_{M}\right)$$
where [*S*] is the glucose concentration, *V_max_* is the highest reaction velocity obtained, and *V_[S]_* and *K_M_* are as explained above.

Apyrase measurements were performed using a 5 nM apyrase solution in a 0.1 M MES buffer with pH 6.0, containing 5 mM CaCl_2_.

## 3. Results

### 3.1. The dn/dc Value

All standard curves showed a linear relationship (*R*^2^ > 0.99) between signal and concentration. As reported previously, the *dn*/*dc* values were found to be 0.145 mL/g and 0.156 RIU/(g/mL) for glucose and for G-6-P, respectively [8]. The value for glucose was in good agreement with other studies reporting values for glucose, ranging from 0.142 to 0.145 RIU/(g/mL), depending on temperature and wavelength [21,22]. The *dn*/*dc* values for ATP and ADP were found to be 0.144 RIU/(g/mL) and 0.163 RIU/(g/mL), respectively. The results are summarized, including the molar values, in Table 1. Based on these findings, an *a priori* assumption was that the enzymatic phosphorylation of glucose produces an increase in RI, whereas the conversion from ATP to ADP should produce a decrease in RI. Thus, the total signal from the enzymatic phosphorylation of 1 millimole of glucose, including the equimolar bisubstrate conversion of ATP to ADP, should yield a total change in RI equal to (4.07−2.61) + (6.95−7.95) = 0.45 × 10^−5^ RIU.

### 3.2. Enzymatic Phosphoryalation of Glucose by Hexokinase

The measured signal from the enzymatic phosphorylation of glucose catalyzed by hexokinase, as seen in Figure 1a, displayed an initial linear increase in the signal, and the slope of this initial linear period was identical for the higher concentrations within the first minute, indicating zero order kinetics (substrate concentrations >> K_M_). Eventually, as the glucose substrate was catalyzed to G-6-P, the signal rate decreased until a maximum plateau (end-point) was reached and the reaction was complete, in agreement with the general Michaelis–Menten kinetics. The measurements were started after 40 s of injection time and, therefore, we lack the signal generated in this time period. Because the reaction proceeded at maximum velocity, this initial lacking signal is linear, and is essentially a constant. Using backwards linear extrapolation on the data, as shown in Figure 1a, we found that this lacking signal was relatively small, equaling 0.02 and 0.03 radians for the smallest and the highest substrate concentrations, respectively. The relationship between the end-point signal and initial glucose concentration was linear, with a slope of 0.99 × 10^−5^ RIU/mM, as seen in Figure 1b.

### 3.3. K_M_ Hexokinase

To determine the K_M_ for hexokinase, the initial velocities from the experiments with glucose concentrations ranging from 0.1 to 2.0 mM were plotted against the substrate concentrations (Figure 2) and K_M_ was found to be 0.33 mM (95% CI: 0.19–0.47).

### 3.4. Enzymatic Hydrolosis of ATP and ADP by Apyrase

The enzymatic hydrolysis of ATP and ADP by apyrase, as seen in the examples shown in Figure 3a, was in accordance with enzymatic reactions, showing an initial linear increase where the reaction proceeded at *V_max_*, and a final plateau with endpoint values that showed a linear relationship when plotted against the initial substrate concentration, as seen in Figure 3b. The estimated signal that was lost during injection (the initial 40 s) was less than 0.01 rad and was disregarded. Surprisingly, this reaction showed a marked increase in the RI, which was contrary to the expected decrease based on the *dn*/*dc* values of ATP and ADP. From the linear regression, the ΔRI was found to be 1.51 × 10^−5^ for ATP and 0.68 × 10^−5^ for ADP. As is discussed below, the reason for this increase in RI could be attributed to the release of free inorganic phosphate (Pi) from the enzymatic hydrolysis of ATP and ADP.

## 4. Discussion

Our results from the enzymatic reactions showed changes of the RI in the order of 10^−5^, well above the RI detection limit of our set-up (7 × 10^−7^ ) [19] for substrate concentrations in the milimolar range. We also found a linear relationship between the substrate and endpoint signal, as seen in Figure 1b, which implies that it would be possible to determine unknown concentrations of glucose from enzymatic reactions using BSI. 

The *K_M_* for hexokinase was found to be 0.33 mM (95% CI: 0.19–0.47), which was in reasonable accordance with the published values of 0.13–0.24 mM [2,4], with variations attributed to experimental conditions such as ionic strength. The relatively large uncertainty in *K_M_* was caused by the limitations of the current experimental setup. For low substrate concentrations ([S]<*K_M_*), a significant amount of substrate will be catalyzed during injection (40 s) and the enzyme does not operate at *V_max_*, which will effectively cause a slight overestimation of K_M_. However, our results demonstrated that refractive index sensing can be used in the estimation of enzyme kinetics, although the system should be further optimized to handle low substrate concentrations.

### Possible Effect of Ions

The enzymatic hydrolysis of ATP performed in our experiments resulted in a large increase in the refractive index upon the hydrolyzation of ATP, which suggests that the inorganic phosphate (Pi) released by the hydrolysis of the γ- and β-phosphate groups of adenosine phosphates made a significant contribution to the refractive index. The refractive index of a solution containing ions is largely dependent on the polarizability of the ions, and there are several studies on this issue [23,24,25,26]. Marcus found the following polarizabilities for ions in solution: Ca^2+^ 0.63 and PO_4_^3−^ 5.99 (× 10^−24^ cm^3^) [24]. Phosphate especially stands out, with a relatively large polarizability, which supports our findings that the release of P_i_ made a significant contribution to the RI of a solution. Although equimolar amounts of AMP are produced from both ATP and ADP as a substrate, two phosphate groups (γ- and β-phosphate) are released from ATP, but only one (β-phosphate) is released from ADP. However, the end-point values for ATP hydrolysis are more than twice as high as those for ADP (1.51 × 10^−5^ vs. 0.68 × 10^−5^), which suggests that the hydrolysis of the γ-phosphate group produces a larger increment in the RI than the hydrolysis of the β-phosphate. This difference in the RI contribution from the phosphate groups could be caused by an inequivalent dissociation of divalent ions chelated with adenosine phosphates. The Ca^2+^ and Mg^2+^ stability constants for the complexes with ATP, ADP, AMP were log(*K_Mg_*) 4.0, 3.0 and 1.6; and log(*K_Ca_*) 3.5, 2.9 and 1.5, respectively [27,28,29]. Thus, a larger amount of cations were dissociated upon β-hydrolysis compared to γ-hydrolysis, which may explain the disproportionate change in refractive index between β- and γ-hydrolysis.

As ATP and ADP are only commercially available as salts, the standard solutions used to determine the *dn*/dc value contained dissociated ions that may affect the refractive index significantly and the experimental determined *dn*/*dc* values for ATP and ADP needs further validation. The theoretical polarizabilities of the adenine phosphates are ATP 35.6, ADP 31.7 and AMP 27.7 (×10^−24^ cm^3^) [30], which is in accordance with our results that showed that the *dn*/*dc* value for ATP was larger than for ADP.

In the case of the enzymatic phosphorylation of glucose by hexokinase, we also found a discrepancy between the increase in the RI measured from the enzymatic reaction (end-point value: 0.99 × 10^−5^ RIU/mM) and the increase estimated from the *dn*/*dc* value (0.45 × 10^−5^ RIU/mM), which further suggests that the *dn*/*dc* value of substrates and of products cannot accurately predict the RI change from enzymatic reactions.

## 5. Conclusions

We demonstrated that the measurement of RI changes can be used to directly measure selected enzymatic reactions. Specifically, this was achieved using BSI with small sample volumes contained in a simple glass capillary. The change in RI was proportional to the formation of the products and could be used to determine the substrate concentrations and to obtain enzymatic kinetic parameters, but the effect of the various components in the solution on the bulk RI was complex and requires further study.

## Figures and Tables

**Figure 1 sensors-19-00539-f001:**
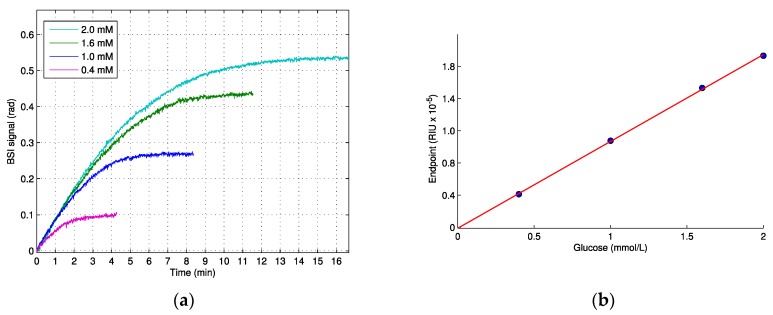
The enzymatic phosphorylation of glucose to glucose-6-phosphate by hexokinase, measured using BSI. (**a**) The real-time measurement of four glucose solutions (0.4, 1.0, 1.6 and 2.0 mM). (**b**) The end-point values in refractive index units vs. initial glucose concentration. Linear fit (*R*^2^ > 0.99, slope 0.99 × 10^−5^ RIU/mM). Figure 1a is reproduced from reference [8] with permission from Elsevier.

**Figure 2 sensors-19-00539-f002:**
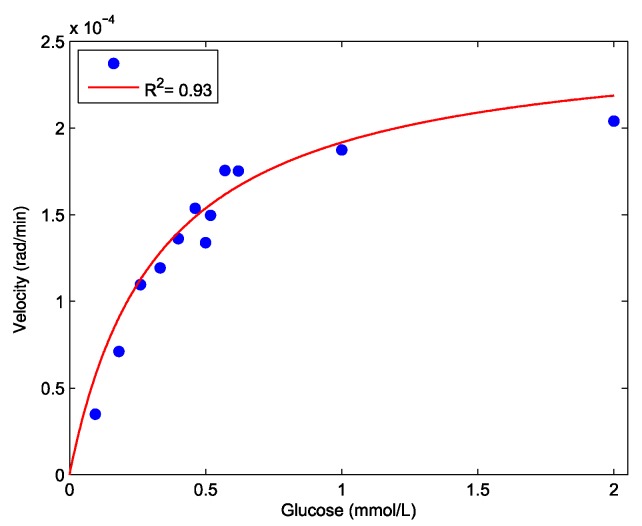
Hexokinase substrate saturation curve: Initial reaction velocity (rad/min) plotted against initial glucose concentrations (mM). Red line shows non-linear regression (*R*^2^ = 0.93) to the parameters in Equation 3 from which K_M_ is obtained.

**Figure 3 sensors-19-00539-f003:**
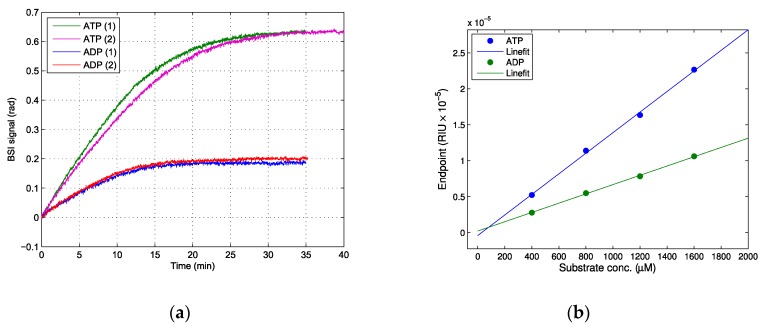
The enzymatic hydrolysis of ATP and ADP to AMP by apyrase. (**a**) The real-time measurement of 1.6 mM ATP and ADP (duplicate runs are shown for both ATP and ADP). (**b**) The linear fit of end-point values in refractive index units vs. initial substrate concentration. ATP (blue) *R*^2^ > 0.99, slope 1.51 × 10^−5^ RIU/mM, ADP (green) R^2^ > 0.99, slope 0.68 × 10^−5^ RIU/mM.

**Table 1 sensors-19-00539-t001:** Summarized findings of the *dn*/*dc* value and the slope of the endpoint values vs. substrate concentration. n is the number of solutions in each standard curve and substrate solutions in the enzymatic reactions.

**Substance**	**n**	***dn*/*dc* (RIU/mM) × 10^−5^**	***dn*/*dc* RIU/(g/mL)^−1^ × 10^−5^**
Glucose	9	2.61	0.145
G-6-P	9	4.07	0.156
ATP	7	7.95	0.144
ADP	7	6.95	0.163
**Enzymatic Reaction**		**Slope of Endpoint (RIU/mM) × 10^−5^**	
Glu→G-6-P(ATP→ADP)	4	0.99	
ATP→AMP+2P_i_	4	1.51	
ADP→AMP+P_i_	4	0.68

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
