# Peer review of "Real-Time Interferometric Refractive Index Change Measurement for the Direct Detection of Enzymatic Reactions and the Determination of Enzyme Kinetics"

_sensors, 2019, doi:10.3390/s19030539_

Round 1
Reviewer 1 Report
This is a review of the manuscript in preparation by Jepsen at al “Real time interferometric refractive index change 2 measurement for direct detection of enzymatic 3 reactions and determination of enzyme kinetics”. Authors are investigating the utility of previously reported BSI configuration to monitor several types of enzymatic reactions that result in a significant RI changes of the solutions. The chosen reaction were enzymatic phosphorylation of D-glucose to glucose-6-phosphate by hexokinase and the enzymatic hydrolysation of adenosine phosphates by apyrase. Manuscript is reasonably well written and requires only several cosmetic changes. My recommendation – Accept
Minor changes:
1. Section 2. Please include probe volume of the setup. Also indicate limit-of-detection (LOD) in RUI and concentration.
2. Lines 124-132. To maintain consistency throughout the manuscript and ease of reading I recommend changing dn/dc units to 0.145 RIU/(g/mL) and 0.156 RIU/(g/mL)...
3. Table 1. I assume “n” in second column means number of experiments. I recommend to spell it out in order to differentiate from “n” as refractive index notation in adjacent columns “dn/dc”.
Author Response
We have corrected the manuscript in accordance with the minor changes suggested:
1. Section 2. Please include probe volume of the setup. Also indicate limit-of-detection (LOD) in RUI and concentration.
We have included both probe volume and LOD to the revised manuscript and referenced to our previous work where we perform a detailed investigation of the systems LOD.
2. Lines 124-132. To maintain consistency throughout the manuscript and ease of reading I recommend changing dn/dc units to 0.145 RIU/(g/mL) and 0.156 RIU/(g/mL)...
Conventionally the dn/dc is reported in ml/g, but we agree that it improves the ease of reading if units of RIU are used consistently, so we have corrected the units accordingly.
3. Table 1. I assume “n” in second column means number of experiments. I recommend to spell it out in order to differentiate from “n” as refractive index notation in adjacent columns “dn/dc”.
The 'n', which has now been clarified in the table caption, describes the number of solutions for each standard curve or enzymatic reaction. The reproducibility of the system has been addressed in previous work (Jepsen et al. Evaluation of back scatter interferometry, a method for detecting protein binding in solution. Analyst 00:1–7. doi: 10.1039/c4an01129e) and for some of the standard curves experiments have been repeated using a conventional deflection refractometer (Jørgensen et al. (2015) Back scattering interferometry revisited – A theoretical and experimental investigation. Sensors Actuators B Chem 220:1328–1337. doi: 10.1016/j.snb.2015.06.121) Both studies are referenced within the current manuscript.
Reviewer 2 Report
In line 132 .you write "..(4.07-2.61)+(7695-7.95)=0.45x10 -5 .. " I´ld like to suggest not using letter x as multiplying symbol, similar line 147
In caption fig.2 it might be helpful to include information about nonlinear model function, as you already provided in text,
line 116: "...Michaelis-Menten constant (K M ) is derived by non-linear parameter estimation from the equation:
V[s] = V max x ([s]/([s]+K M )) ..."
Author Response
The suggestions from this reviewer have been implemented fully in the revised manuscrpt. We thank the reviewer for spotting the formatting error using "x" as multiplication sign, this was an omission on our part.
In regards to line 116.. the non-linear parameter estimation of KM. We have clarified the section regarding the estimation of KM in manuscript section 2.3. Furthermore the figure caption has been revised to reflect these changes.
Reviewer 3 Report
This research article introduces Back Scatter Interferometry (BSI) as a sensitive method for monitoring the conversion of glucose to glucose-6-phosphate, as well as conversion of ATP/ADP to AMP. The conclusion is strongly supported by the experiments result and the initial reaction velocities KM is measured to be 0.33 mM. Overall, the study demonstrates that changes in refractive index can be used for direct determination of substrate concentrations and enzyme kinetics.
Overall the article is well-structured and the results are convincing, but I have a few questions and suggestions below:
1. It is not very clear to me the novelty of this article. The novelty of the research should be pointed out clearly in the abstract and further elaborated in the introduction section around the previous work. I suggest author further work on this part.
2. Line 81: The experiment setup should be more clearly introduced in this article, including the model of the instruments and chemicals.
3. Line 178 mentioned there is a linear relationship between substrate and endpoint signal. In what range does this linear relationship holds true? If substrate concentration further increase, would this relationship become higher order?
4. Are those experiments repeatable in lab and can be reproduced in other labs?
Author Response
Thank you for the very relevant questions/suggestions: Given the limited time available we are not able to fully comply with all your suggestions but have adressed your points as follows:
Ad 1: We have restructured the introduction to emphazise that using differential RI measurements on bulk solutions for monitoring enzymatic reactions is a novel unexplored area. The introduction now includes more references to other studies where enzymatic reactions have been examined using solid surface refractive index sensors and describe briefly how our method differs. To the best of our knowledge, we demonstrate for the first time how bulk refractive index changes of solution can be used to detect the enzymatic reaction of Apyrase and derive KM of Hexokinase.
The abstract has been altered so the sentence now reads: (with underlining showing changes)
Back Scatter Interferometry (BSI) is a sensitive method for detecting changes in the bulk refractive index of a solution in a microfluidic system. Here we demonstrate that BSI can be used to directly detect enzymatic reactions and for the first time derive kinetic parameters
Ad 2: We have not described the detail of the set-up as they are given in ref 8 and 19, which we refer to. A detailed description with figures would not be possible withing the 5 days time given to submit a revised manuscript.
Ad 3: This is a good question that however, is outside the scope of the current paper as it will call for further studies not possible to conduct within the current time frame. Other papers on BSI have addressed the linearity of the system, including (Swinney K, Markov D, Bornhop DJ (2000) Chip-scale universal detection based on backscatter interferometry. Anal Chem 72:2690–2695. doi: 10.1021/ac000261r)
Ad 4: With the results presented we highly suggest other researchers/labs to perform similar studies, not necessarily using BSI, but other kinds of differential RI measurements. We have previously reproduced both the dn/dc of glucose and G-6-P and the hexokinase experiment using the commercial T-Rex instrumentation (ref 8)
Sincerely
Søren Jepsen and co-authors